# VIDEOCANVAS: UNIFIED VIDEO COMPLETION FROM ARBITRARY SPATIOTEMPORAL PATCHES VIA IN-CONTEXT CONDITIONING

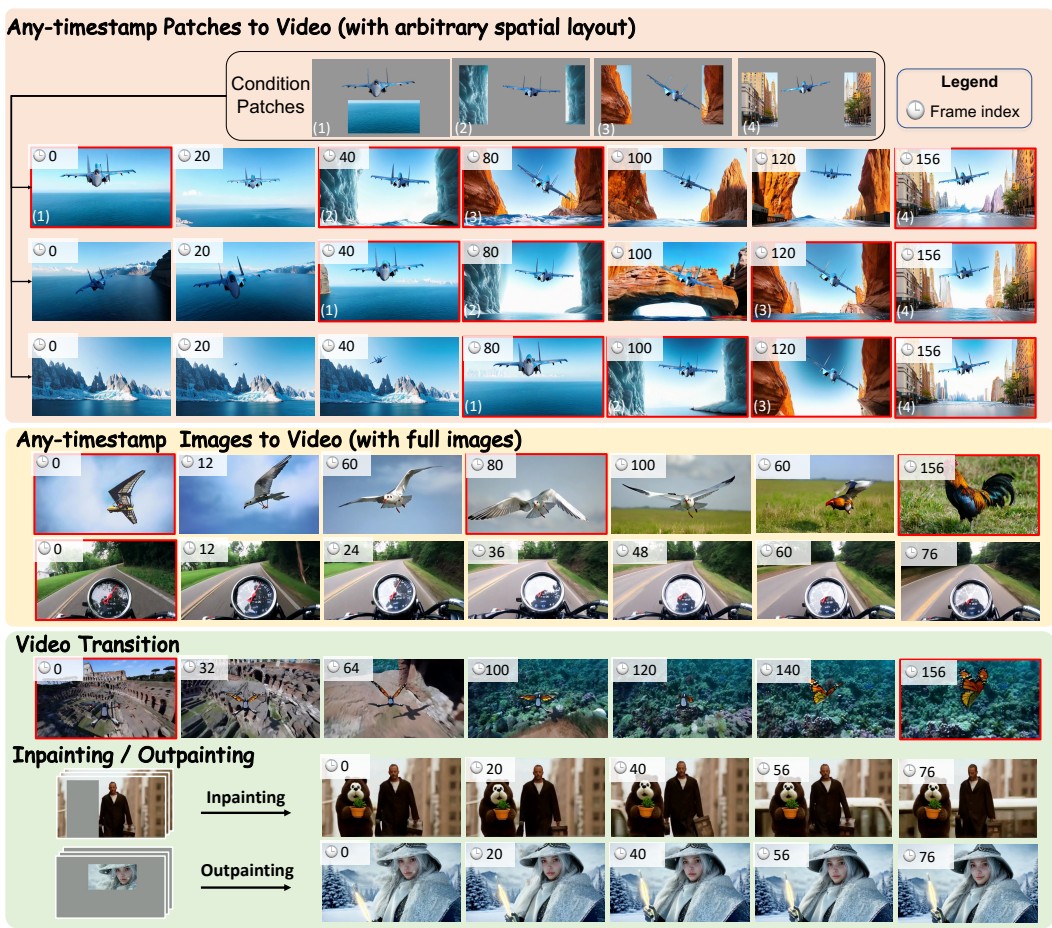

Figure 1: **VideoCanvas: Arbitrary Spatio-Temporal Video Completion.** Given any conditions (frames or patches, outlined in red), the model fills in the remaining gray regions to generate coherent, high-quality videos. This unified formulation subsumes various tasks such as Any-Frame/Patch-to-Video, inpainting, outpainting, and cross-scene video transitions, all in a zero-shot manner. More results are available on our anonymous project page or supplementary project page. **Best viewed zoomed in.**

## ABSTRACT

We introduce the task of arbitrary spatio-temporal video completion, where a video is generated from arbitrary, user-specified patches placed at any spatial location and timestamp, akin to painting on a *video canvas*. This flexible formulation naturally unifies many existing controllable video generation tasks—including first-frame image-to-video, inpainting, extension, and interpolation—under a single, cohesive paradigm. Realizing this vision, however, faces a fundamental obstacle in modern latent video diffusion models: the temporal ambiguity introduced

by causal VAEs, where multiple pixel frames are compressed into a single latent representation, making precise frame-level conditioning structurally difficult. We address this challenge with **VideoCanvas**, a novel framework that adapts the In-Context Conditioning (ICC) paradigm to this fine-grained control task with zero new parameters. We propose a hybrid conditioning strategy that decouples spatial and temporal control: spatial placement is handled via zero-padding, while temporal alignment is achieved through Temporal RoPE Interpolation, which assigns each condition a continuous fractional position within the latent sequence. This resolves the VAE's temporal ambiguity and enables pixel-frame-aware control on a frozen backbone. To evaluate this new capability, we develop VideoCanvas-Bench, the first benchmark for arbitrary spatio-temporal video completion, covering both intra-scene fidelity and inter-scene creativity. Experiments demonstrate that VideoCanvas significantly outperforms existing conditioning paradigms, establishing a new state of the art in flexible and unified video generation.

# 1 INTRODUCTION

Video generation has made significant strides with the advent of Diffusion Transformers (DiTs) (Peebles & Xie, 2023; Chen et al., 2023; Wang et al., 2025a; Yang et al., 2024), marking a turning point in the field's ability to synthesize high-quality videos. However, generating videos that truly align with user intent remains a significant challenge. Existing controllable approaches are typically constrained by rigid, task-specific formats—for example, conditioning only on a first frame (Guo et al., 2023; Kong et al., 2024), using an initial clip with limited temporal horizon (Bar et al., 2025; Yang et al., 2025a), or performing structural inpainting and outpainting (Zhou et al., 2023; Wang et al., 2024; Yang et al., 2025b). These methods treat spatio-temporal control as a set of isolated problems, lacking a unified approach. We propose a unified approach to bridge these fragmented tasks: *treating video synthesis as painting on a spatio-temporal canvas*. In this framework, users can place arbitrary content patches at any location and timestamp, and the model will synthesize a complete, temporally consistent video around them, as illustrated in Fig. 1. This fine-grained control enables a wide range of applications, from creative content generation to practical use cases, such as reconstructing videos from partially transmitted or corrupted data packets (Li et al., 2023; Du et al., 2020), or generating videos with specific spatial and temporal conditions for diverse domains.

Realizing this vision presents fundamental challenges across both spatial and temporal dimensions, inherent to modern latent video models. Temporally, causal video VAEs compress multiple pixel frames into a single latent slot, creating indexing ambiguity and making frame-accurate control a core obstacle, as illustrated in Fig. 2(a). Spatially, conditions may take arbitrary forms—from full frames to small, irregular patches—requiring a mechanism that can seamlessly unify inpainting and outpainting within one formulation. The core difficulty lies in designing a conditioning paradigm that can resolve both temporal ambiguity and spatial irregularity simultaneously.

Viewed through this lens, the limitations of existing paradigms become clear. Latent Replacement (HaCohen et al., 2024; Kong et al., 2024) was designed mainly for first-frame I2V but fails to generalize, as it overwrites entire latent slots and disrupts temporal consistency once applied to arbitrary timestamps. Channel Concatenation and Adapter-style injection methods (Yang et al., 2024; Wang et al., 2025a; Mou et al., 2024; Zhang et al., 2023) fuse conditional features either by concatenating at the input or injecting via lightweight encoders. Despite architectural differences, these approaches remain coarse-grained: pixel-frame-aware control ultimately requires feeding zero-padded frames to the VAE, but pretrained VAEs are not robust to such inputs. Making them work would require expensive VAE fine-tuning and re-training of the DiT backbone. More recent In-Context Conditioning (ICC) methods (Tan et al., 2024; Ju et al., 2025; He et al., 2025; Ye et al., 2025a) inherit the same difficulty when naively combined with zero-padding: they still demand VAE/DiT re-training to handle the distribution shift, and further double the sequence length by encoding padded frames, resulting in severe inefficiency during both training and inference.

In this paper, we introduce **VideoCanvas**, the first framework to apply *In-Context Conditioning* to the challenging task of arbitrary spatio-temporal video completion. We also propose a hybrid conditioning strategy that decouples space and time: spatial alignment is achieved by zero-padded

Figure 2: **Core challenge and solution for pixel-frame-aware conditioning.** **(a)** Causal VAEs create temporal ambiguity by mapping frames to a single latent. We propose a hybrid solution combining Spatial Padding with Temporal RoPE Interpolation. **(b)** We show how competing paradigms are ill-suited for fine-grained control, while our *ICC* approach provides an effective solution.

VAE encoding of arbitrary patches, while temporal ambiguity is resolved by our novel RoPE Interpolation, which assigns continuous fractional indices to conditional frame tokens. This design removes the need for costly re-training of the VAE or architectural modifications of the DiT backbone, while allowing efficient fine-tuning to enable fine-grained pixel-frame-aware control within a simple, parameter-free ICC architecture.

To evaluate this new task and framework, we present **VideoCanvasBench**, a comprehensive benchmark tailored for arbitrary spatio-temporal video completion. To the best of our knowledge, it is the first to systematically incorporate multi-frame, non-homologous image and patch conditions to test both intra-scene fidelity and inter-scene creativity. Our contributions are as follows:

- We introduce and formalize the task of arbitrary spatio-temporal video completion, a unified framework that encompasses a wide range of controllable video generation scenarios, including not only existing first-frame-to-video, video extension and painting tasks, but also new tasks such as any-timestamp patch-to-video and any-timestamp image-to-video, extending control to arbitrary timestamps in time and arbitrary regions in space.

- We propose VideoCanvas, the first framework to apply the *In-Context Conditioning* paradigm to the task of arbitrary spatio-temporal completion. We further introduce a hybrid conditioning strategy: Spatial Zero-Padding and Temporal RoPE Interpolation. This approach enables efficient fine-tuning of the DiT model without the need for VAE retraining, achieving fine-grained spatiotemporal control.

- We design and release VideoCanvasBench, the first benchmark explicitly dedicated to arbitrary spatio-temporal completion, and demonstrate that VideoCanvas achieves state-of-the-art performance across diverse settings, outperforming existing conditioning paradigms.

## 2 RELATED WORK

### 2.1 ARBITRARY SPATIO-TEMPORAL VIDEO COMPLETION

Controllable video generation aims to synthesize content that adheres to user inputs beyond a simple text prompt. Existing approaches are often constrained by rigid, task-specific formats, such as conditioning only on a first frame (Guo et al., 2023; Kong et al., 2024; Wan et al., 2025; Shi et al., 2024; Gao et al., 2025), on a short initial sequence (Bar et al., 2025; Yang et al., 2025a), or on structural inpainting and outpainting (Zhou et al., 2023; Wang et al., 2024; Bian et al., 2025; Yang et al., 2025b). Conceptually, these represent special cases of the broader challenge of video completion, yet prior work has treated them as separate sub-tasks, each requiring specialized solutions. Recent unified frameworks like VACE (Jiang et al., 2025) have made progress on consolidating diverse tasks, primarily focusing on inpainting, outpainting, and video extension. However, these models still remain constrained to specific forms of video completion and fail to address the general problem of arbitrary spatio-temporal control.

In contrast, we introduce and formalize the task of *arbitrary spatio-temporal video completion*, a unified and flexible paradigm that subsumes and extends prior settings. By allowing conditions of arbitrary shapes and at arbitrary spatio-temporal locations, our task formulation goes beyond task-specific or partially unified approaches. This enables genuinely unified spatio-temporal video generation. To facilitate systematic evaluation of this capability, we also introduce *VideoCanvas-Bench*, the first benchmark designed specifically for this setting, and demonstrate how our approach outperforms existing methods across a range of video completion tasks.

## 2.2 Paradigms for Video Conditioning

Tackling the flexible task of arbitrary spatio-temporal completion requires a robust and fine-grained conditioning mechanism. Existing paradigms (Shown in Fig. 2) for injecting condition into a video foundation model can be broadly categorized as follows.

**Latent Replacement.** This paradigm, used in LTX-Video and Hunyuan-Video (HaCohen et al., 2024; Kong et al., 2024), was primarily designed for the *first-frame image-to-video* setting. In this case, overwriting the initial latent with an encoded image remains relatively compatible with the training distribution. However, extending it to arbitrary frames introduces a clear train–inference mismatch: temporal VAEs compress multiple frames into one latent slot during training, while inference substitutes that slot with a single-frame latent. This inconsistency often disrupts temporal dynamics, leading to static frames or motion collapse.

**Channel Concatenation and Adapter-based Injection.** A straightforward strategy is to fuse conditional features into the model's data stream at fixed locations. Recent I2V models such as CogVideoX (Yang et al., 2024) and Wan (Wang et al., 2025a) adopt this variant by concatenating condition and noisy latents along the channel axis at the input layer. Extensions like T2I-Adapter (Mou et al., 2024), VACE (Jiang et al., 2025) and ControlNet (Zhang et al., 2023) process conditions through lightweight encoders before injecting them into intermediate layers. Despite their differences, these approaches share a fundamental limitation: pixel-frame-aware control requires feeding zero-padded frames to the VAE, but pre-trained VAEs are not robust to such out-of-distribution inputs. Making them work would require costly VAE fine-tuning and retraining of the DiT backbone, which is prohibitively expensive for our task.

**Cross-Attention Injection.** Another line of work adds cross-attention layers to incorporate conditioning features, typically for global controls such as text, audio, or style cues (Cui et al., 2025; Meng et al., 2025; Blattmann et al., 2023; Ye et al., 2025b). While effective for holistic guidance, this design requires substantial architectural modifications and introduces many new parameters, limiting scalability.

**In-Context Conditioning (ICC).** ICC, a paradigm pioneered in the image domain by OminiControl (Tan et al., 2024) and extended to video by FullDiT (Ju et al., 2025; He et al., 2025) and UNIC (Ye et al., 2025a), represents a unified, parameter-free conditioning approach. It treats all inputs, including content and conditions, as tokens within a single sequence, processed jointly by self-attention. This simple yet effective design allows for flexible conditioning, but ICC still struggles with the key challenge of *pixel-frame ambiguity* introduced by causal VAEs, which makes precise temporal alignment difficult.

Building on ICC, we are the first to adapt this paradigm to the task of arbitrary spatio-temporal video completion. We introduce a Hybrid Conditioning Strategy, which decouples spatial and temporal challenges. The novel alignment strategy, Temporal RoPE Interpolation, enables pixel-frame-aware conditioning on frozen causal VAEs, unlocking ICC's full potential for this setting.

## 3 Methodology

### 3.1 Task Definition and Problem Setup

We introduce the task of **arbitrary spatio-temporal video completion**, a unified formulation that generalizes and subsumes a wide range of controllable video generation scenarios. Formally, let a video be denoted as $X = \{x_0, x_1, \ldots, x_{T-1}\}$ with $T$ frames. A user provides a set of spatio-temporal conditions $\mathcal{P} = \{(p_i, m_i, t_i)\}_{i=1}^{M}$, where $p_i$ is a patch (full-frame or partial), $m_i$ is a

Figure 3: **The pipeline of VideoCanvas**, which fine-tunes a base T2V model for arbitrary spatio-temporal control with zero new parameters. Our framework leverages the In-Context Conditioning (ICC) paradigm. After preparing conditional patches with zero-padding for spatial placement, we use independent VAE encoding for temporal decoupling. Our RoPE Interpolation then aligns each discrete token by mapping its source pixel-frame index $Y$ to a fractional position $Y/N$, where $N$ is the VAE temporal stride (here, $N = 4$). As illustrated, this maps Frame 41 to position 10.25. This strategy enables fine-grained control without architectural changes.

spatial mask specifying its placement within a frame, $t_i \in [0, T-1]$ is the temporal index of target frame, and $M$ denotes the total number of conditions provided by the user. The goal is to generate a coherent video $\hat{X}$ such that

$$\hat{X}[t_i] \odot m_i \approx p_i, \quad \forall i \in \{1, \dots, M\},$$

while simultaneously completing all unconditioned regions with plausible and consistent content.

This task naturally unifies many prior settings as special cases: *image-to-video* (when $\mathcal{P}$ contains the first full frame), *video extension* (when $\mathcal{P}$ provides the first video clip), *video inpainting/outpainting* (when $\mathcal{P}$ contains masked regions in video frames), and *interpolation* (when $\mathcal{P}$ specifies keyframes at first and last timestamps). By allowing arbitrary spatial masks at arbitrary timestamps, our definition goes strictly beyond these rigid formats, enabling a single framework to address them all.

### 3.2 PRELIMINARIES

**Video DiT with 3D RoPE.** Our work builds upon a latent video diffusion model that uses a Diffusion Transformer (DiT) backbone (Peebles & Xie, 2023) and is trained with a modern flow matching objective (Lipman et al., 2022). Crucially, to handle the spatio-temporal nature of video data, the model's self-attention mechanism is equipped with 3D Rotary Positional Embeddings (RoPE) (Su et al., 2024).

The 3D RoPE integrates both temporal and spatial information by applying a rotation to the query and key vectors in the attention mechanism. This rotation is driven by the position of each token in the 3D space (time, height, and width), allowing the model to capture both temporal continuity and spatial relationships. This mechanism is central to our ability to perform precise temporal alignment and interpolation in video generation tasks.

**Hybrid Video VAE.** Modern video foundation models employ a *Hybrid Video VAE* (Zhao et al., 2024; Yang et al., 2024; Wu et al., 2025) that supports both image and video modes. In video mode, the encoder performs causal temporal compression with a fixed stride $N$ (e.g., $N = 4$). At the beginning of a sequence, frames are replicated so that the first latent uniquely corresponds to the first pixel frame. Subsequently, every $N$ consecutive frames collapse into a single latent slot. Formally, for a pixel-frame index $i$, the latent index is

$$\text{latent\_idx}(i) = \left\lceil \frac{i}{N} \right\rceil,$$

with replication ensuring that $i = 0$ maps to latent index 0. This stride-based compression is computationally efficient but introduces a pixel-frame ambiguity: multiple frames (e.g., Frame 1 and Frame 3) may collapse to the same latent, making precise frame-level conditioning non-trivial.

## 3.3 VIDEOCANVAS PIPELINE

To address the challenge of arbitrary spatio-temporal completion, we propose VideoCanvas, a unified framework built upon the In-Context Conditioning (ICC) paradigm. We are the first to leverage ICC for this task, introducing a novel *hybrid conditioning strategy* that decouples spatial and temporal alignment, enabling fine-grained, pixel-frame-aware control on a frozen VAE and a fine-tuned DiT with zero new parameters. The entire pipeline is illustrated in Fig. 3.

**Spatial Conditioning via Zero-Padding.** As shown on the left of Fig. 3, our process begins at the pixel level. For each conditional patch, we construct a full-frame canvas, place the patch in its correct spatial location, and fill the remaining pixels with zeros. This preserves the absolute positional information required for spatial control.

**Temporal Decoupling via Independent VAE Encoding.** Next, each of these zero-padded frames is encoded *independently* by the frozen VAE in its image mode. This is a critical step for temporal decoupling: by encoding each frame individually, we bypass the VAE's causal temporal compression mechanism. The result is a set of conditional latent tokens, $\mathbf{x}_P$, where each token purely represents its corresponding single pixel frame, free from the temporal ambiguity discussed in our preliminaries.

**Temporal Alignment via RoPE Interpolation.** The final and most crucial step is to precisely align these decoupled conditional tokens within the DiT's 3D spatio-temporal grid. We leverage the continuous nature of the 3D RoPE used by our DiT backbone. For a conditional token originating from a pixel frame with index $Y$, we assign it a fractional temporal position $t = Y/N$, where $N$ is the VAE's temporal stride.

As illustrated in the center of Fig. 3, this maps a condition from Frame 41 to the fractional temporal coordinate $t = 10.25$. When this token is processed by the DiT, its query and key vectors are rotated by an angle $\theta$ that is a function of this fractional position $(10.25, h, w)$. This allows the self-attention mechanism to understand that this condition should exert its influence at a point in time *between* the 10th and 11th integer latent slots. This RoPE Interpolation strategy is what enables the precise, sub-latent, pixel-frame-aware temporal control that is structurally inaccessible to other paradigms.

**Unified Sequence Denoising.** Finally, the prepared conditional tokens $\mathbf{x}_P$ (with their fractional positions) and the standard noisy latent tokens $\mathbf{x}_t$ (with their integer positions) are concatenated into a single sequence. This unified sequence is then processed by the DiT, which is fine-tuned under the standard flow-matching objective to denoise the sequence and generate the final video.

## 4 VIDEOCANVASBENCH

Existing benchmarks focus on rigid tasks such as I2V or outpainting, and cannot assess the flexible spatio-temporal control central to our formulation. We therefore introduce *VideoCanvasBench*, the first benchmark systematically designed for arbitrary spatio-temporal video completion.

The benchmark probes two complementary capabilities: high-fidelity completion within a single scene (homologous) and creative synthesis across different sources (non-homologous). It consists of three categories: (1) **Patch-Level**, using partial patches at fixed anchor timestamps (start, middle, end). We construct all seven possible combinations—single-frame (S, M, E), two-frame (S+M, S+E, M+E), and three-frame (S+M+E)—to evaluate interpolation fidelity under varying temporal sparsity. (2) **Image-Level**, using full-frame conditions at the same timestamps, designed to test the completion of full-frame content. (3) **Video-Level**, covering video-level completion scenarios such as inpainting, outpainting, and transitions between non-homologous clips. In total, VideoCanvasBench comprises over **2,000** test cases. Further construction details are provided in Appendix C.

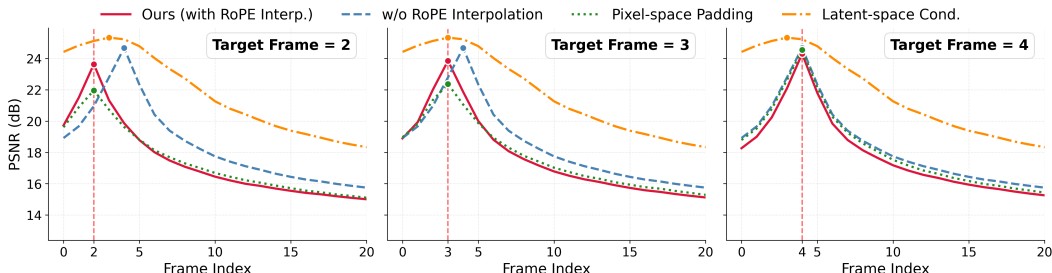

Figure 4: **Impact of Temporal RoPE Interpolation.** Per-frame PSNR for single-frame I2V with targets 2/3/4. Our method (red, solid) peaks exactly at the target frame. "w/o RoPE Interpolation" (blue, dashed) misaligns, "Latent-space Condition" (orange, dot-dashed) collapses motion, and "Pixel-space Padding" (green, dotted) is precise but degraded.

## 5 EXPERIMENTS

Our experiments are designed to answer two central questions: (1) Can our proposed *Temporal RoPE Interpolation* resolve the temporal ambiguity of causal VAEs, thereby enabling precise pixel-frame alignment beyond the native VAE stride? (2) Even under the coarse granularity imposed by latent slots, is the *In-Context Conditioning (ICC)* paradigm intrinsically more effective than prior mechanisms such as Latent Replacement and Channel Concatenation? We address the first question through an ablation study of different pixel-frame alignment strategies (Sec. 5.3), and the second via a paradigm-level comparison on our benchmark (Sec. 5.4).

### 5.1 SETUP

**Backbone and Training.** We build our framework upon an internal latent video diffusion model, as no existing open-source model is designed for our new task of arbitrary spatio-temporal completion (see Appendix A for details). The model is fine-tuned for 20k steps on 650k video clips ($384 \times 672$ resolution, 5 seconds) using the Adam optimizer with a learning rate of $5 \times 10^{-5}$ and a batch size of 32 on 32 GPUs. Inference uses 50 DDIM steps with a CFG scale of 7.5.

**Baselines.** As our task is new, no existing work provides a direct solution. For fair comparison, we compare three representative conditioning paradigms (Fig. 2b) on the same backbone: (1) *Latent Replacement*, as used in LTX-Video and HunyuanVideo (HaCohen et al., 2024; Kong et al., 2024), (2) *Channel Concatenation*, widely adopted in CogVideoX and Wan (Yang et al., 2024; Wang et al., 2025a), and (3) our *In-Context Conditioning (ICC)*. All paradigms are trained under identical settings and constrained to the same set of conditionable frames defined by the VAE stride, ensuring a rigorous and controlled comparison. More details are shown in the appendix.

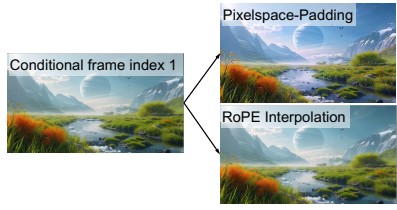

Figure 5: Padding vs. RoPE Interp.

| Method | PSNR↑ | Dynamic Degree↑ | Imaging Quality↑ |
|---|---|---|---|
| Ours (RoPE Interp.) | 23.86 | **39.75** | **71.61** |
| w/o RoPE Interp. | 22.95 | 23.00 | 70.85 |
| Pixel-space Pad. | 22.02 | 30.25 | 71.50 |
| Latent-space Cond. | **25.13** | 5.00 | 71.17 |

Table 1: Ablation on single-frame I2V

### 5.2 EVALUATION METRICS

**Automated Metrics.** Fidelity is measured by PSNR and FVD (Unterthiner et al., 2018), and perceptual quality by four metrics: Aesthetic Quality (LAION-AI, 2022), Imaging Quality (Ke et al., 2021), Temporal Coherence (Cai et al., 2025), and Dynamic Degree (Teed & Deng, 2020).

**User Study.** To complement automated metrics, we conducted a user study with 25 participants on 30 randomly sampled cases. In each case, participants viewed side-by-side outputs from three meth-

ods in a three-way forced-choice setting, and rated them along three axes: Visual Quality (quality and dynamics), Semantic Quality (faithfulness to text and images), and Overall Preference (holistic choice). Results are reported as win rates (%) over competing methods.

## 5.3 ABLATION STUDY: PIXEL-FRAME ALIGNMENT STRATEGIES

As discussed in Fig. 2(a), causal video VAEs map several pixel frames into one latent, which creates ambiguity when conditioning on a specific frame. One intuitive workaround is to keep the target frame and pad the rest with zeros before VAE encoding, which we denote as *Pixel-space Padding*. While this approach is temporally precise, it forces the frozen VAE to process highly out-of-distribution inputs, often corrupting colors and textures. To disentangle this issue, we compare four alignment strategies: (i) *Latent-space Conditioning*: encode the entire video with the VAE (video mode) to obtain a latent sequence; at designated timestamps, the corresponding latent slice is injected as the conditional input. (ii) *Pixel-space Padding*: construct a pixel-space video in which non-target frames are zeroed; encode the entire padded video with the VAE (video mode) (iii) *w/o RoPE Interpolation*: encode each conditional frame independently with the VAE (image mode); assign each conditional token to the discrete temporal slot determined by the VAE compression window (no interpolation). (iv) *Our full method with Temporal RoPE Interpolation*.

**Qualitative evidence.** Although pixel-space padding can in principle "point" to the correct frame, it introduces visible artifacts because the VAE never trained on zero-filled inputs. Fig. 5 illustrates this: the padded result shows clear color shifts and texture wash-out, whereas RoPE-based alignment preserves the conditional frame with high fidelity.

**Quantitative analysis.** We further evaluate single-frame I2V at target indices (2, 3, 4). As shown in Fig. 4 and Tab. 1, *Latent-space Conditioning* yields a nearly flat PSNR curve, indicating motion collapse. *w/o RoPE Interpolation* recovers dynamics but shifts the PSNR peaks due to slot misalignment. *Pixel-space Padding* peaks at the correct indices but with lower overall fidelity. In contrast, our method with RoPE Interpolation aligns exactly to the target frames and achieves the best fidelity.

Together, these results make two points clear. First, padding-based solutions, despite being temporally precise, degrade quality due to VAE signal corruption. Second, latent-space conditioning and integer-only alignment cannot resolve frame-level ambiguity. In contrast, our ICC with Temporal RoPE Interpolation uniquely provides both fine-grained control and high-fidelity generation.

## 5.4 MAIN RESULTS: PARADIGM COMPARISON

Having established that padding-based solutions are impractical due to quality degradation, we next compare the three representative conditioning paradigms—*Latent Replacement*, *Channel Concatenation*, and our *In-Context Conditioning*—under identical settings, where each latent corresponds to a pixel frame. This ensures that performance differences arise solely from the conditioning mechanism itself (not zero-padding).

**Quantitative Comparison.** Tab. 2 shows results on VideoCanvasBench across three task categories: Patch Level, Image Level and Video Level. The data reveals a consistent trend across all task categories. **Latent Replacement** achieves deceptively high scores in static similarity metrics like PSNR, but at the cost of synthesizing motion. Its extremely low Dynamic Degree scores indicate that it generates nearly frozen videos, which is reflected in its poor FVD, confirming a large distributional gap with real videos. **Channel Concatenation** produces more dynamics but consistently lags behind our method in both reference fidelity (PSNR, FVD) and key perceptual metrics. In contrast, our **ICC** strikes the best balance, achieving competitive fidelity while attaining the highest Dynamic Degree. Most importantly, the **User Study** confirms ICC's superiority, where it is overwhelmingly preferred by human evaluators across all three task levels.

**Qualitative Comparison.** Fig. 6 illustrates representative cases. In the two-frame I2V task (Fig. 6a), Latent Replacement collapses into static repetition around the conditioning frame, while Channel Concatenation introduces unnatural distortions in the deer's body. ICC instead generates smooth and plausible motion while maintaining identity. In the more challenging two-frame P2V setting (Fig. 6b), the weaknesses of the baselines become even clearer. Latent Replacement produces abrupt, unnatural transitions, and Channel Concatenation suffers from severe identity drift, with

Table 2: Performance comparison on our VideoCanvasBench across three task categories. The best result is in **bold** and the second best is underscored. Higher scores are better for all metrics (except FVD). Abbreviations: AQ = Aesthetic Quality, IQ = Imaging Quality, TC = Temporal Coherence, DD = Dynamic Degree, VQ = Visual Quality, SQ = Semantic Quality, OP = Overall Preference.

| Method | Reference Fidelity | | Perceptual Metrics(%) | | | | User Study(%) | | |
|---|---|---|---|---|---|---|---|---|---|
| | PSNR↑ | FVD↓ | AQ↑ | IQ↑ | TC↑ | DD↑ | VQ↑ | SQ↑ | OP↑ |
| **Patch Level** (First Frame, First-Last Frames and Any Keyframes) | | | | | | | | | |
| Replace. (Kong et al., 2024) | **24.29** | 19.335 | 55.45 | **69.19** | **91.04** | 21.00 | 14.62 | 16.15 | 14.23 |
| Channel. (Yang et al., 2024) | 23.73 | 18.147 | **55.54** | 68.49 | 89.36 | 39.44 | 26.54 | 26.92 | 25.38 |
| ICC (Ours) | 23.83 | **17.553** | 55.53 | 68.87 | 89.71 | **40.44** | **58.85** | **56.92** | **60.38** |
| **Image Level** (First Frame, First-Last Frames and Any Keyframes) | | | | | | | | | |
| Replace. (Kong et al., 2024) | **26.72** | 12.534 | **55.64** | **69.32** | **90.37** | 24.22 | 8.46 | 7.31 | 7.31 |
| Channel. (Yang et al., 2024) | 25.83 | 10.947 | 55.37 | 68.74 | 88.88 | 41.22 | 23.46 | 27.31 | 24.23 |
| ICC (Ours) | 26.06 | **10.805** | 55.40 | 69.25 | 89.02 | **44.78** | **68.08** | **65.38** | **68.46** |
| **Video Level** (Video Transition, Video Inpainting and Outpainting) | | | | | | | | | |
| Replace. (Kong et al., 2024) | **23.90** | 15.958 | 53.28 | 67.23 | 89.37 | 47.39 | 5.00 | 4.23 | 5.00 |
| Channel. (Yang et al., 2024) | 23.54 | 11.371 | **54.16** | 68.33 | 88.88 | 53.04 | 26.92 | 25.77 | 25.38 |
| ICC (Ours) | 23.68 | **10.252** | 53.16 | **69.43** | **89.40** | **53.20** | **68.08** | **70.00** | **69.62** |

(a) Images to Video: First-Middle Frame    (b) Patches to Video: First-Last Frame

Figure 6: **Qualitative comparison.** Our method preserves high quality and smooth transitions.

a kangaroo inexplicably mutating into a dog mid-video. ICC alone preserves motion, identity, and structural consistency throughout the sequence, avoiding both freezing and semantic corruption.

Overall, both quantitative and qualitative evidence converge on the same conclusion. Our ablation study (Sec. 5.3) demonstrates that Temporal RoPE Interpolation uniquely enables fine-grained pixel-frame alignment without sacrificing fidelity, while the paradigm comparison (Sec. 5.4) shows that even at coarse latent-level granularity, ICC consistently outperforms Latent Replacement and Channel Concatenation. Taken together, these findings establish ICC as the most robust and effective conditioning mechanism for arbitrary spatio-temporal video generation.

# 6 CONCLUSION

We introduced and formalized the task of arbitrary spatio-temporal video completion. To tackle the core challenge of temporal ambiguity in causal VAEs, we proposed VideoCanvas, a framework based on In-Context Conditioning. We also propose a hybrid conditioning strategy combining Spatial Zero-Padding and Temporal RoPE Interpolation, first to enable fine-grained, pixel-frame-aware control on a frozen VAE via efficient DiT fine-tuning. Experiments on our new benchmark, VideoCanvasBench, confirm the superiority of our approach, establishing a robust and generalizable foundation for the future of controllable video synthesis.

## ETHICS STATEMENT

**Scope and Intended Use.** **VideoCanvas** is designed for the task of **arbitrary spatio-temporal video completion**, intended for research, education, and creative prototyping. Its core function is to enable users to synthesize a complete video from a sparse set of user-provided spatio-temporal patches. It is *not* intended for surveillance, impersonation of real individuals, political persuasion, or other high-risk deployments where generated content could be used to deceive or harm. We will accompany any artifact release with a research-only license and an acceptable-use policy (AUP) that explicitly prohibits such abusive or unlawful scenarios.

**Misuse Risks and Mitigations.** Like any powerful generative model, VideoCanvascarries potential risks of misuse. These include the creation of "deepfake" content for identity impersonation, targeted harassment, deceptive political messaging, and the generation of pornographic, violent, or otherwise harmful media. Our mitigations to curb these risks include: (i) a research-only release of our model and code to the academic community; (ii) leveraging default content filters inherited from the base model to block the generation of clearly harmful categories (e.g., sexual content, explicit violence, hate symbols).

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

## APPENDIX

## A   INTRODUCTION OF THE BASE TEXT-TO-VIDEO GENERATION MODEL

We use an internal transformer-based latent diffusion model (Peebles & Xie, 2023) as the base T2V generation model, as illustrated in Fig. S7. We employ a 3D-VAE to transform videos from the pixel space to a latent space, upon which we construct a transformer-based video diffusion model. Unlike previous models that rely on UNets or transformers, which typically incorporate an additional 1D temporal attention module for video generation, such spatially-temporally separated designs do not yield optimal results. We replace the 1D temporal attention with 3D self-attention, enabling the model to effectively perceive and process spatiotemporal tokens, thereby achieving a high-quality and coherent video generation model. Specifically, before each attention or feed-forward network (FFN) module, we map the timestep to a scale, thereby applying RMSNorm to the spatiotemporal tokens.

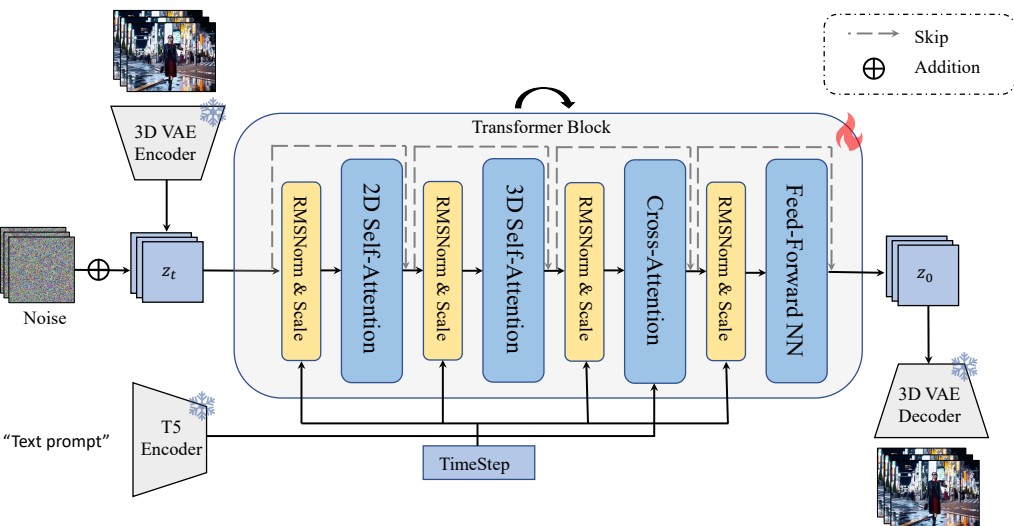

Figure S7: **Overview of the base text-to-video generation model.**

## B   IMPLEMENTATION DETAILS

**Conditioning paradigms.**   Since no existing work is trained for our new task of arbitrary spatio-temporal completion, we re-implement three representative paradigms on the same base model for fair comparison (Fig. 2b), following the references used in the main text:

- **Latent Replacement** (HaCohen et al., 2024; Kong et al., 2024). For a given conditional frame, the corresponding latent tokens are overwritten with VAE-encoded ground-truth latents. Training applies a masked loss only to non-conditional regions, while conditional regions are assigned timestep 0.

- **Channel Concatenation** (Yang et al., 2024; Wang et al., 2025a). Condition frames are encoded into latents, assembled into a zero-padded latent sequence, and concatenated with the noisy latent sequence along the channel dimension. A learnable projection layer then restores the embedding dimension. In our implementation, concatenation is applied *after patchification*, as this setting empirically yields the best results; applying it before patchification leads to degraded visual quality. The tradeoff is that after-patchify concatenation substantially increases the channel dimensionality, resulting in a projection layer with ∼16.6M trainable parameters. Thus, while this design enriches the conditioning signal and improves learning, it comes at the cost of significantly more parameters compared to the other paradigms.

- **In-Context Conditioning (ICC)** (Tan et al., 2024; Ju et al., 2025). Our method encodes condition frames into clean latent tokens and concatenates them with the noisy sequence along the token

dimension. Temporal alignment is achieved with our RoPE Interpolation strategy (Sec. **??**). The loss is applied only to noisy tokens, while conditional tokens are assigned timestep 0. This design requires no additional trainable parameters.

All paradigms are trained under identical settings and restricted to the same set of conditionable frames defined by the VAE stride, ensuring a rigorous and controlled comparison.

**Temporal granularity.** Different conditioning paradigms impose different constraints on the indices where conditional frames can be applied. As discussed in Fig. 2 and Sec. 5.3, both *Channel Concatenation* and *Latent Replacement* require zero-padded inputs in order to achieve pixel-frame-aware control, which significantly degrades quality. To ensure a fair comparison across paradigms, we therefore restrict all methods to a coarser temporal granularity. Specifically, with a VAE temporal stride of 4, the conditionable frame indices are standardized to the discrete set $\{0, 4, 8, \ldots, 76\}$. This guarantees that each method is trained on exactly the same set of indices and receives comparable supervision.

**Training strategy.** At each iteration, three frames are randomly sampled from a source video to serve as temporal anchors. From each anchor frame, we extract a spatial region by cropping a patch covering between 20%–100% of the original frame size. This unified training strategy ensures that the model encounters a diverse spectrum of conditioning scenarios, ranging from sparse local patches to nearly complete frames, and from early anchors to late anchors. Such exposure allows the model to learn arbitrary spatio-temporal conditioning in a single framework.

### B.1 DETAILED EVALUATION METRICS

This section provides additional details of the evaluation metrics used in our experiments. As described in the main paper, we evaluate video generation quality using *PSNR*, *FVD*, *Aesthetic Quality*, *Imaging Quality*, *Temporal Coherence*, and *Dynamic Degree*, together with a complementary user study. Our protocol is designed to measure two aspects: (1) fidelity to visual conditions when ground-truth is available, and (2) perceptual quality and temporal consistency in general cases.

**Reference-based Visual Fidelity.** We use two complementary metrics to evaluate reference-based visual fidelity:

- **PSNR**: We adopt PSNR to evaluate reconstruction accuracy in the conditional regions, measuring how faithfully the generated outputs reproduce the provided visual inputs. This metric focuses on pixel-level reconstruction accuracy and is widely used for assessing the quality of generated images and videos.

- **Fréchet Video Distance (FVD)** (Unterthiner et al., 2018): We also employ FVD to measure the distance between the distributions of generated and real video sequences, capturing both temporal and spatial information. A lower FVD indicates higher similarity between the generated video and real videos, reflecting better overall quality. This metric is particularly useful for comparing video generation models by assessing their ability to match the distribution of real-world videos. However, FVD is only applicable to the parts of our dataset with ground-truth videos (i.e., homologous videos). For non-homologous image-to-video and video transition tasks, where no ground truth exists, we do not compute FVD.

**Perceptual Quality and Consistency Metrics.** We further employ the following metrics to comprehensively assess perceptual quality and temporal behavior:

- **Aesthetic Quality** (LAION-AI, 2022): evaluates the artistic and aesthetic value of each frame using the LAION aesthetic predictor. It reflects high-level properties such as composition, color harmony, photo-realism, and naturalness.

- **Imaging Quality** (Ke et al., 2021): measures the absence of low-level distortions using the MUSIQ predictor. It is sensitive to degradations such as over-exposure, noise, and blur, thereby reflecting frame-level fidelity from a perceptual perspective.

- **Temporal Coherence** (Cai et al., 2025): evaluates temporal stability by computing CLIP feature similarity (CSCV) between *adjacent* frames. This is a modification of the Background Consistency metric from VBench (Huang et al., 2024), which compared all frames to the first frame. That design fails for videos with significant camera motion or large scene transition, where the first frame is not a valid reference. This adjacent-frame-only formulation provides a more robust measure of local temporal smoothness.
- **Dynamic Degree** (Teed & Deng, 2020): quantifies the level of motion by estimating optical flow magnitudes with RAFT. This prevents models from trivially achieving high consistency with static outputs, and explicitly rewards natural, dynamic motion.

**User Study.** To complement automated metrics, we conducted a user study with 25 participants on 30 randomly sampled cases. Each case contained outputs from three methods, presented in a three-way forced-choice setting. Participants rated results along three axes: (1) *Visual Quality* (quality and dynamics), (2) *Semantic Quality* (faithfulness to text and images), and (3) *Overall Preference* (holistic choice). We report results as win rates (%) over competing methods.

## C  VIDEOCANVASBENCH CONSTRUCTION DETAILS

This section provides a comprehensive overview of the data curation and task generation pipeline for *VideoCanvasBench*, the first systematic evaluation suite for arbitrary spatio-temporal video completion.

### C.1  DATA CURATION

We curate two complementary types of sources: (1) *homologous* videos for testing fidelity within a single coherent scene, and (2) *non-homologous* images and videos for evaluating creativity across distinct content.

**Homologous Video Set (100 Videos).** We began with an initial pool of ∼2,000 videos from Pexels (Pexels, 2025). A multi-stage filtering pipeline was applied to ensure quality and diversity:

- Blur filtering: blurry videos were removed by calculating the CV2.Laplacian (Bradski, 2000) score for each frame and excluding those below a threshold of 200.
- Motion filtering: static or nearly-static clips were excluded using RAFT-based motion magnitude thresholds exceeding 5 (Teed & Deng, 2020).
- Length filtering: only videos longer than 5 seconds were retained.

From this pool, we selected 100 diverse, high-quality clips covering a wide range of scenes (e.g., human activities, animals, landscapes). All were standardized to 77 frames at 15 FPS to provide a consistent evaluation length. Each video is paired with captions generated by a captioning model fine-tuned on Koala36M (Wang et al., 2025b) following the LLaVA-based (Liu et al., 2023) annotation pipeline. All captions are further verified by human annotators to ensure accuracy in both content and motion descriptions.

**Non-Homologous Image and Video Sets.** To test the ability to synthesize across unrelated contexts, we manually curated visually distinct sources from Pexels (Pexels, 2025) and Unsplash (Unsplash, 2025), ensuring large appearance and semantic gaps. The set includes:

- 50 pairs of non-homologous images, selected to maximize dissimilarity (e.g., indoor vs. outdoor, object vs. scene).
- 50 triplets of non-homologous images, further increasing combinatorial diversity.
- 30 pairs of non-homologous video clips, curated for challenging video transitions, similar to the blending function of Sora (OpenAI, 2023).

These non-homologous cases explicitly test the model's capacity for creative interpolation and cross-scene reasoning. Each non-homologous source is annotated with captions automatically generated by Gemini 2.5 Pro (Comanici et al., 2025) and manually corrected to ensure faithful descriptions of both appearance and motion.

## C.2 Benchmark Task Definitions

**Task 1: Image-Level (Any-Timestamp Image-to-Video).** This task uses full frames as conditions to test temporal reasoning and interpolation fidelity. We explicitly construct nine sub-tasks by combining conditions from fixed temporal anchors: start (frame 1), middle (frame 41), and end (frame 77).

- Homologous cases. From each source video we sample three anchor frames (start, middle, end), and construct:
  - *Single-frame I2V:* start → video, middle → video, end → video.
  - *Two-frame I2V:* start+end → video, start+middle → video, middle+end → video.
  - *Three-frame I2V:* start+middle+end → video.
- Non-homologous cases. For curated pairs of images, we construct the three two-frame tasks (start+end, start+middle, middle+end). For curated triplets of images, we construct the three-frame task (start+middle+end). Each non-homologous source is annotated with captions automatically generated by Gemini 2.5 Pro (Comanici et al., 2025) and manually checked for accuracy.

**Task 2: Patch-Level (Any-Timestamp Patch-to-Video).** This variant follows the same nine sub-task definitions as Image-Level setting, but replaces each full-frame condition with a cropped patch.

- Patch extraction. For each conditional frame, patches are obtained via a semi-automated process: 50% object-aware masks using SAM (Kirillov et al., 2023) or YOLO (Ultralytics, 2023), and 50% random crops.
- Temporal anchors. The same start, middle, and end frame positions are used to construct single-, two-, and three-frame variants, for both homologous and non-homologous cases.
- Difficulty. The subset explicitly includes challenging cases with very small subjects, requiring the model to extrapolate from minimal context.

**Task 3: Video-Level (Transition, Inpainting and Outpainting).** This task evaluates more general video-level completion scenarios beyond frame- or patch-level control. It consists of three sub-categories:

- *Video Transition.* For 30 curated pairs of non-homologous video clips, the first clip provides the start segment and the second the end segment, while the model synthesizes the intermediate transition. This setup parallels the blending function explored in Sora (OpenAI, 2023). Each case is annotated with captions generated by Gemini 2.5 Pro (Comanici et al., 2025) and manually corrected to ensure faithful descriptions of both content and motion.
- *Inpainting.* For homologous videos, interior rectangular masks are applied to each frame, covering 20%–50% of the width/height. The model must fill the missing regions with temporally consistent content.
- *Outpainting.* Boundary masks are applied to crop the central region, masking out 60%–90% of the width/height. The model is required to extrapolate plausible outer regions beyond the visible content.

## C.3 Scale

In total, *VideoCanvasBench* includes over **2,000** test cases: 900 for Patch-Level, 900 for Image-Level, and 230 for Video-Level. Each case is designed to probe a specific aspect of fidelity, creativity, or temporal reasoning in the proposed unified task.

## C.4 Licensing and Annotations.

All videos in our benchmark are sourced from Pexels (Pexels, 2025), and images are sourced from both Pexels and Unsplash (Unsplash, 2025). Content on Pexels is provided under the Pexels License, which permits free use for commercial and non-commercial purposes without requiring attribution, with restrictions against reselling unaltered copies, use in trademarks, or misuse of identifiable people or brands. A subset of Pexels content is explicitly marked as Creative Commons Zero (CC0),

which places the work in the public domain. Unsplash photos are provided under the Unsplash License, which similarly allows free commercial and non-commercial use without attribution, while prohibiting resale of unaltered content, creation of competing stock services, or misleading association with brands or people. In both cases, all curated data is legally licensed for academic research use.

Captions generated by Gemini 2.5 Pro (Comanici et al., 2025) were manually verified by the authors to ensure accuracy and consistency across all benchmark cases.

# D  ADDITIONAL ANALYSIS OF ZERO-PADDED INPUTS

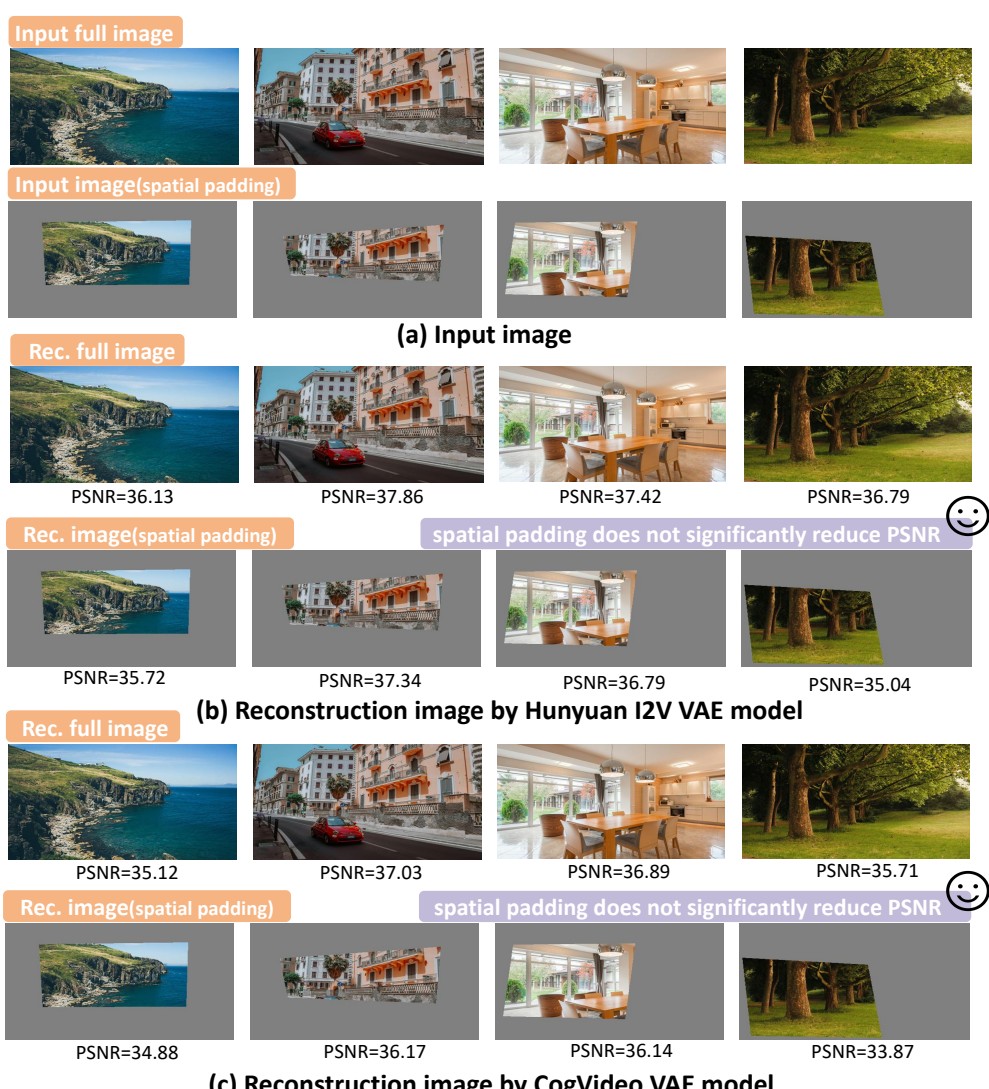

Figure S8: **Robustness of Hybrid Video VAEs to Spatial Padding**. This figure demonstrates that both the Hunyuan I2V and CogVideo VAE models can tolerate spatial zero-padding well. When reconstructing images with large zero-padded regions (middle row), the PSNR values are only slightly lower than those of the full, unpadded images (top row). Crucially, the original content within the non-zero regions is faithfully preserved, while the padded areas remain visually neutral. This empirical evidence confirms that our spatial conditioning strategy, which relies on zero-padding before VAE encoding, is stable and practical, enabling precise spatial control without degrading the quality of the conditioned content.

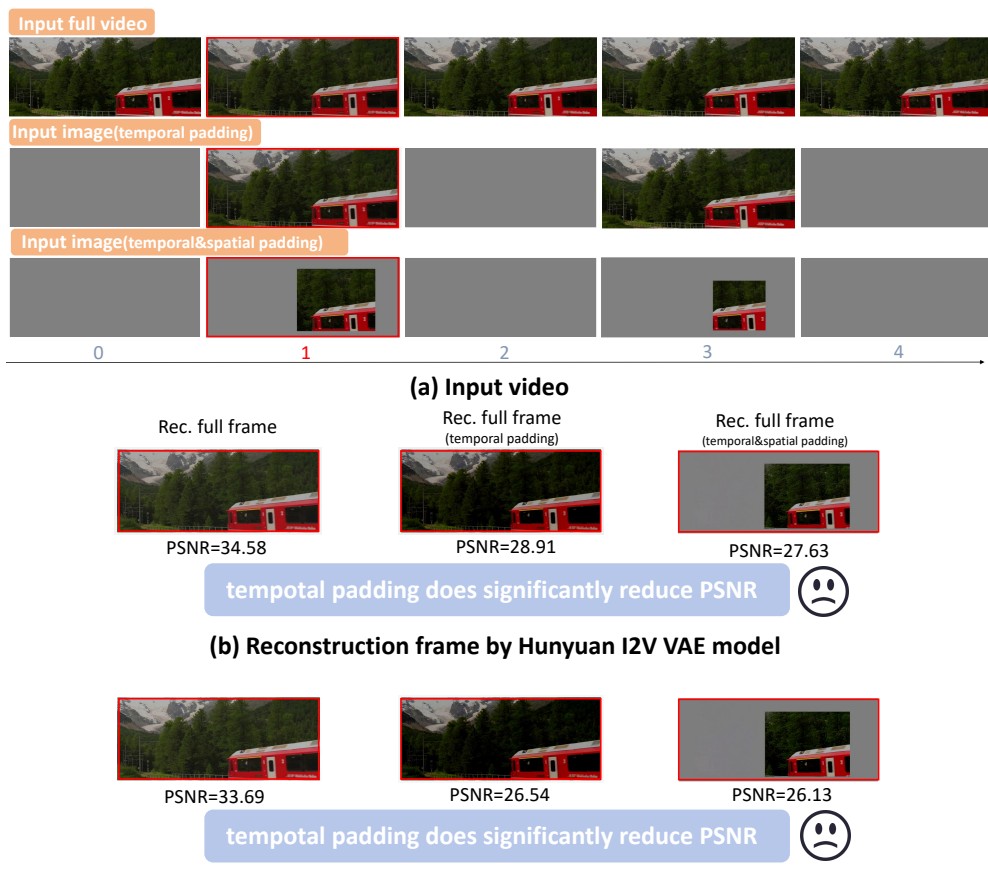

(a) Input video

(b) Reconstruction frame by Hunyuan I2V VAE model

(c) Reconstruction frame by CogVideo VAE model

Figure S9: **Vulnerability of Hybrid Video VAEs to Temporal Padding**. This figure contrasts the robustness observed in spatial padding. When applying temporal zero-padding (where only specific frames contain content), both VAE models suffer a relatively great drop in reconstruction quality. The PSNR values for the padded reconstructions (bottom rows) are much lower than those of the full video (top row), demonstrating a degradation in fidelity. The reconstructed frames exhibit noticeable color shifts, and loss of detail, highlighting that the VAE cannot handle such distributionally mismatched inputs. This mode underscores why direct temporal zero-padding is ineffective and validates the necessity of our Temporal RoPE Interpolation strategy, which avoids this problem by operating at the latent token level with fractional positions.

In Section 3, we describe using zero-padding to indicate unconditioned regions when preparing conditional frames. This approach is crucial for our spatial conditioning strategy, as it allows us to precisely specify the location of a condition patch within a frame without modifying the pre-trained VAE backbone. However, a critical question arises: can a standard hybrid video VAE, trained on natural images and videos, effectively handle inputs that contain large areas of zero-valued pixels (i.e., spatial padding)? As illustrated in Figure S8 and Figure S9, this distinction between spatial and temporal padding is fundamental to understanding our method.

To address this, we conducted an empirical study using two popular pre-trained VAE models: Hunyuan I2V and CogVideo. We evaluated their robustness to both spatial and temporal padding under realistic conditions.

**Setup.** We collected 20 diverse full-resolution images and 20 short video clips from YouTube, representing a wide range of content (e.g., landscapes, cityscapes, indoor scenes, moving vehicles). For each image, we applied random spatial zero-padding masks, covering approximately 40-60% of the pixels. For each video clip, we created three types of padded inputs: 1. A video with conditional

frames containing the original content, while all other frames are filled with zeros (pure temporal padding). 2. A video where conditional frames contains cropped region of the original content, with all other frames being zero (temporal & spatial padding).

Each input was then encoded and decoded using the two hybrid VAE model. We measured the reconstruction fidelity using PSNR and qualitatively inspected the outputs.

**Results.** The results provide clear evidence of the differential impact of padding modes:

Spatial Padding Robustness: As shown in Figure S8, both VAE models demonstrate remarkable tolerance to spatial zero-padding. The average PSNR of reconstructed images with spatial padding is only marginally lower than that of the baseline (full image), with an average drop of **0.89 dB**(Hunyuan I2V) and **1.13 dB**(CogVideo).

Temporal Padding Vulnerability: In stark contrast, Figure S9 reveals the limitations of traditional approaches. When applying temporal zero-padding (encoding a single frame into a sequence where most frames are zero), both VAE models exhibit a dramatic degradation in reconstruction quality. The average PSNR drops by over **6.12 dB**(Hunyuan I2V) and **7.01 dB**(CogVideo) compared to the baseline.

**Conclusion.** These findings confirm that the key to achieving pixel-frame-aware control lies in decoupling spatial and temporal handling. Our method leverages the inherent robustness of the VAE to spatial padding while bypassing the ineffectiveness of temporal padding through our proposed Temporal RoPE Interpolation. This separation allows us to harness the power of a frozen, pre-trained VAE for flexible and high-fidelity video completion, avoiding the need for costly retraining or architectural modifications. The experimental results thus strongly validate the necessity and effectiveness of our approach.

## E    ADDITIONAL ANALYSIS OF TEMPORAL ROPE INTERPOLATION

Figure 4 in the main paper has shown that our Temporal RoPE Interpolation achieves *precise one-to-one alignment* between condition frames and their target temporal positions. Here we further demonstrate that such pixel-frame-level precision is not only feasible, but also *crucial* for improving video completion quality.

To this end, we conduct an additional experiment on the homologous video set (100 videos) from *VideoCanvasBench*. Each video contains 77 frames. We compare two conditioning strategies:

- **Sparse condition:** only the 0th and 4th frames are provided, and the model interpolates the missing frames implicitly.

- **Dense condition:** the 0th–4th frames are explicitly provided, ensuring frame-wise alignment at every step.

Both settings are used to generate the full 77-frame video. We evaluate fidelity by computing PSNR between the generated outputs and the original ground-truth video, focusing on the first 5 frames, and report averages over all 100 videos.

Table R3: Average PSNR (dB) across 100 videos under sparse vs. dense conditioning.

| Condition Type | Conditioned Frames | PSNR ($\uparrow$) |
|---|:---:|:---:|
| Sparse (two frames) | 0, 4 | 24.789 |
| Dense (five frames) | 0, 1, 2, 3, 4 | 25.033 |

The results indicate that explicitly conditioning on consecutive frames yields consistently higher PSNR, demonstrating that RoPE Interpolation not only ensures precise alignment at arbitrary timestamps (as shown in Figure 4) but also effectively leverages *dense temporal cues* to improve reconstruction fidelity. This finding highlights the flexibility of VideoCanvas: it can operate effectively

under sparse conditions for efficient zero-shot completion, while also benefiting from denser conditions that offer stronger temporal guidance, particularly in long-horizon generation where they mitigate motion collapse compared to first-frame-only baselines.

## F  Training and Inference Cost

Tab. R4 summarizes the computational cost of different conditioning paradigms. Our ICC approach introduces **no additional parameters**, whereas channel concatenation requires a projection layer with $\sim$16.6M parameters. Training ICC takes slightly longer (24.5h vs. 21–22h) due to the additional conditioning tokens in the sequence. During inference, our method exhibits a small overhead that grows with the number of conditional frames (168s $\rightarrow$ 175s $\rightarrow$ 184s), because a longer context increases the sequence length processed by the transformer. While this makes inference marginally slower than the baselines with fixed cost, the trade-off is acceptable since ICC consistently achieves the best fidelity and alignment results (see Sec. 5.3, Tab. 1, Fig. 4 and Tab. 2).

Table R4: Training and inference cost comparison across paradigms. Training time is measured over 20k steps. Inference time is per 77-frame video at $384 \times 672$ with different numbers of conditional frames.

| Method | Params | Train | Inference | | |
|---|---|---|---|---|---|
| | | | 1 frame | 2 frame | 3 frame |
| Latent Replacement | 0 | 21.47h | 159s | 159s | 159s |
| Channel Concat | **16.6M** | 22.47h | 164s | 164s | 164s |
| Ours | 0 | **24.54h** | **168s** | **175s** | **184s** |

## G  More Qualitative Results

We provide extended visualizations on all three benchmark tasks defined in Sec. C: (1) Patch Level, (2) Image Level, and (3) Video Level. Figures S11, S10, and S12 showcase side-by-side comparisons with baseline paradigms. Across diverse cases, our ICC with RoPE consistently produces smoother motion, sharper details, and better temporal alignment.

## H  Use of Large Language Models (LLMs)

**Scope of use.**  We used a large language model (LLM) *only for writing polish*, including grammar correction, phrasing refinement, and improvements to clarity and readability. The LLM did *not* contribute to research ideation, problem formulation, method design, experimental setup, result selection, interpretation, or drafting of technical content (theorems, algorithms, proofs, metrics, or analyses). All technical claims, experiments, figures, tables, and conclusions were conceived, implemented, and verified by the authors.

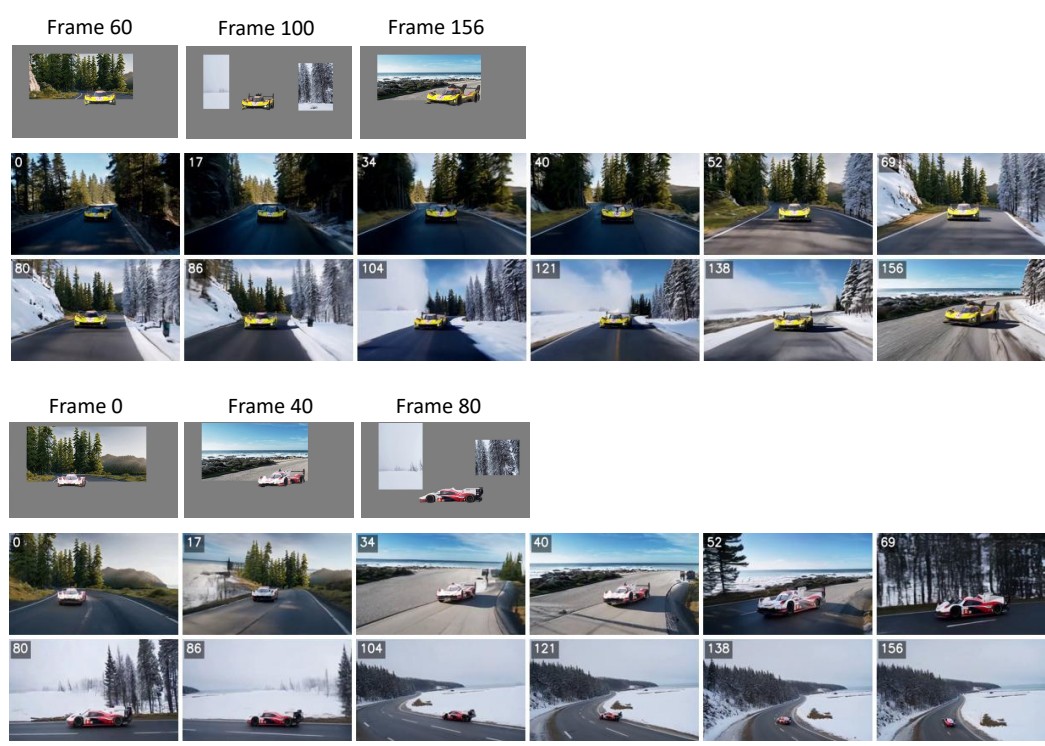

Figure S10: Results on Any-timestamp Patches to Videos

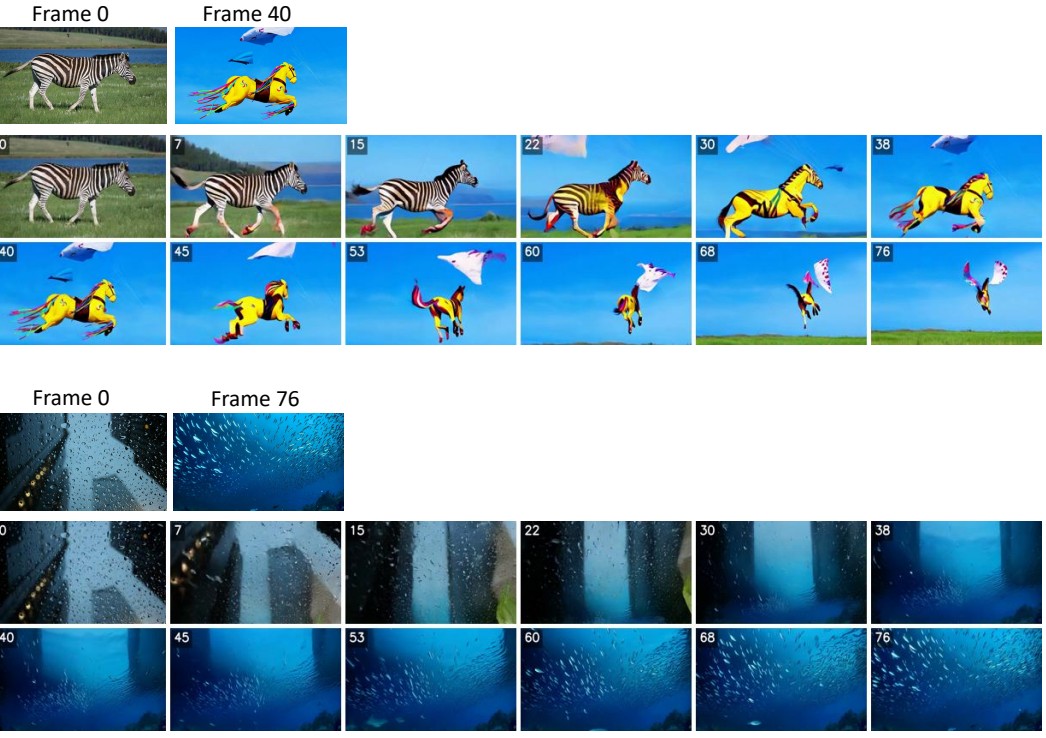

Figure S11: Results on Any-timestamp images to Videos

Source Video 1

Source Video 2

Generated Video

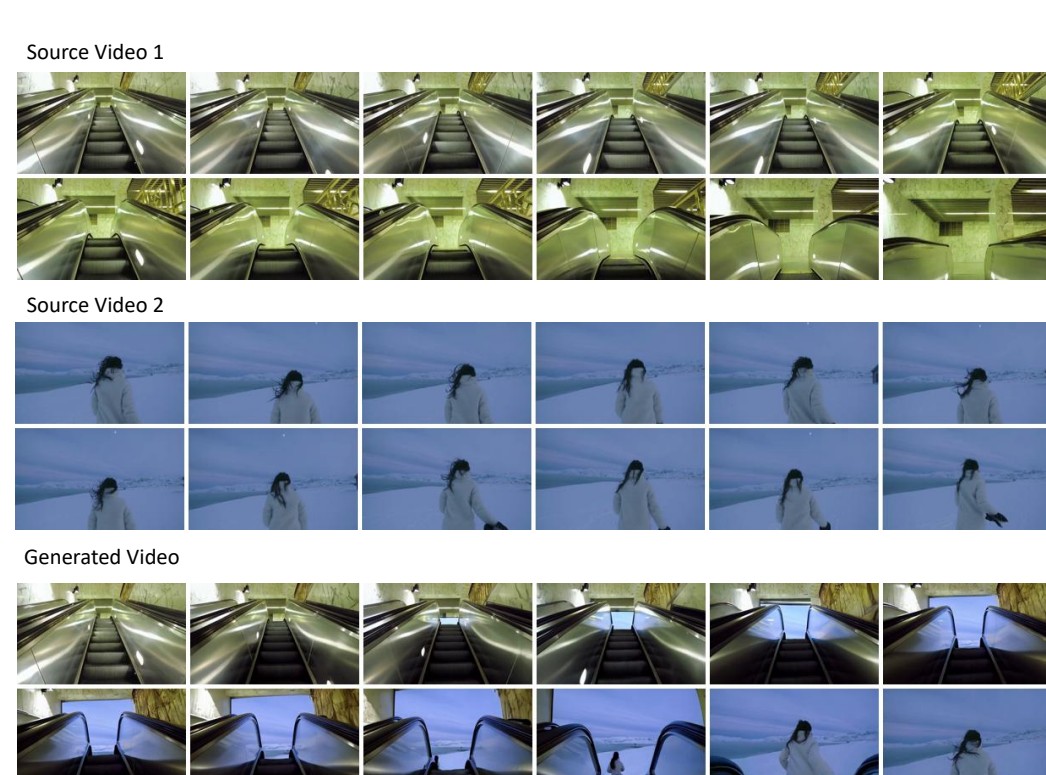

Figure S12: Result on Video transition

