# OpenReview forum: "VideoCanvas: Unified Video Completion from Arbitrary Spatiotemporal Patches via In-Context Conditioning"
_ICLR.cc/2026/Conference — ICLR 2026 Conference Withdrawn Submission_

### Official Review · Reviewer_t3n9 · 2025-10-17

**Soundness:** 3
**Presentation:** 3
**Contribution:** 3
**Rating:** 4
**Confidence:** 4

**Summary:**

The paper introduces a unified task of arbitrary spatio-temporal video completion, which can handle image-to-video, video extension, video inpainting/outpainting, and video interpolation. To accomplish this unified task, the paper proposes VideoCanvas, the first framework to apply the In-Context Conditioning. Further, a comprehensive benchmark called VideoCanvasBench was presented to evaluate this unified task and the VideoCanvas method. The logic behind this presentation is clear. In light of the contributions, the given framework of VideoCanvas is a good contribution, as it can handle a variety of tasks uniformly. The so-called VideoCanvasBench is also a contribution, from the dataset and benchmark perspective.

**Strengths:**

The proposed VideoCanvas is an interesting pipeline, which is applicable to a variety of tasks. The used Temporal Alignment via RoPE Interpolation is different compared to the existing utilization of RoPE.

The paper also presents a VideoCanvasBench benchmark, which can be used to evaluate different tasks under the same experimental settings. This could help attract more researchers in this field.

The paper offers a comprehensive study on the representative conditioning paradigms, including Latent Replacement, Channel Concatenation and the used In-Context Conditioning.

These are the major strengths offered by this paper.

**Weaknesses:**

The paper also shows several weaknesses that require further attention, which will be detailed below.

1. The new task of spatio-temporal video completion is eventually conveyed by the experimental setting in Table 2, where Patch Level, Image Level, and Video Levels are considered. These can be applied to applications of image-to-video, video extension, video inpainting/outpainting, and video interpolation. I am confused by the role of ICC here. What is the deciding factor to unify all the three tasks together? Was this due to the ICC?

2. If the ICC was the main factor to unify all the different tasks, then it looks like the conventional Latent Replacement and Channel Concatenation can also unify these tasks. This is related to the question in 1.

3. The experimental results presented in Table 2 are also not very convincing. I see the ICC achieves the best in the user study scenario. But it is noted that this user study is a subjective experiment. For other metrics such as Reference Fidelity and Perceptual Metrics, the ICC does not show superior performance compared to Latent Replacement and Channel Concatenation. This is the major weakness of this paper, as the proposed ICC does not show great advantages.

4. The paper also mentions that the pre-trained VAEs are not robust to out-of-distribution inputs, such as dealing with zero-padded frames. But when checking the results presented in Figure S8 of the Appendix, the reconstruction result from the CogVideo VAE and Hunyuan I2V VAE does not show much degradation. Could this mean that the pre-trained VAEs are still robust to out-of-distribution inputs? If that is the case, the motivation of this work could also be challenged.

5. The proposed RoPE Interpolation is something interesting, which generates fractional positions, compared to the normally used integer positions. However, the advantage of this RoPE Interpolation is still not very well explained.  Based on Figure 3, there is a mapping between fractional positions and integer positions. That means these two types of positions are interchangeable?

6. The paper first processes the video frames independently, and then unifies all of them via the RoPE Interpolation. Are there any experiments to support such a process to be more useful than directly using 3D VAEs? This is something to be clarified or pointed out, as I do not clearly see the experimental results to demonstrate that.

7. There is a typo in the paper, where our RoPE Interpolation strategy (Sec. ??). This is due to incorrect latex compiling, which needs certain attention.

**Questions:**

I would like the paper to address the highlighted questions here which are related to the weakness section.

1. What is the deciding factor to unify all the three tasks together? What role does ICC play here?

2. Why are the pre-trained VAEs considered not robust to out-of-distribution inputs?

3. What are the clear differences between fractional positions and integer positions? Are they interchangeable?

4. What is the motivation to disentangle the processes of first encoding the frames independently and then unifying them later? Are there any experiments to substantiate this claim?

---

### Official Review · Reviewer_NA7c · 2025-10-29

**Soundness:** 3
**Presentation:** 2
**Contribution:** 2
**Rating:** 6
**Confidence:** 5

**Summary:**

This paper proposes a unified framework that can encompass a wide range of controllable video generation tasks. To achieve this goal, this paper applies the in-context conditioning paradigm and introduces a hybrid conditioning strategy. Furthermore, this paper proposes VideoCanvasBench to evaluate the proposed method.

**Strengths:**

1. The paper is well organized.

2. The topic of unified video generation guided by a single framework is worth exploring in the research community. This paper targets this important problem.

3. The paper provides additional supplementary materials to help reviewers better evaluate the performance of the proposed method.

**Weaknesses:**

There are some concerns and questions about this paper:

1.	Spatial Conditioning via Zero-Padding mentions zero padding at the pixel level, but VAEs are trained on normal images or videos. Can VAEs handle zero-padding images normally?

2.	In the Temporal Decoupling via Independent VAE Encoding section, how many conditional frames are inserted? Are these conditional frames continuous?

3.	In the "Temporal Alignment via RoPE Interpolation" section, the authors emphasize the importance of adding RoPE to the incoming conditional frame. Can the authors visually demonstrate the visual difference between using and not using RoPE? This is because some previous I2V methods, such as Animate Anything [1], do not add positional encoding to the conditional image.

4.	The paper emphasizes that video VAE introduces temporal ambiguity. Can the authors show specific video examples?

5.	The authors mention that this is a new task and therefore lacks a suitable baseline for comparison. However, I believe it is an upgraded version of similar work like SEINE [2] and VDT [3], and cannot be considered a completely new task. Therefore, for patch-level experiments, the authors can compare with VideoBooth [4], for image-level experiments, they can compare with SEINE and ToonCrafter, and for video-level experiments, they can compare with video inpainting/outpainting methods.

**Reference**:

[1] Animateanything: Consistent and controllable animation for video generation, CVPR 2025

[2] Seine: Short-to-long video diffusion model for generative transition and prediction, ICLR 2024

[3] Vdt: General-purpose video diffusion transformers via mask modeling, ICLR 2024

[4] Videobooth: Diffusion-based video generation with image prompts, CVPR 2024

[5] Tooncrafter: Generative cartoon interpolation, TOG 2024

**Questions:**

Please see above. I think it would be better to provide some videos that support Motivation, and also add some comparative experiments.

---

### Official Review · Reviewer_nB97 · 2025-10-29

**Soundness:** 3
**Presentation:** 3
**Contribution:** 2
**Rating:** 4
**Confidence:** 3

**Summary:**

The paper introduces VideoCanvas, a model that utilizes latent video diffusion to perform arbitrary spatio-temporal video completion. The proposed framework is capable of video inpainting and outpainting, conditioned on either a subset of input frames or even specific patches within those frames. The authors also design VideoCanvasBench, a comprehensive benchmark for spatio-temporal video completion encompassing these tasks, which they plan to publicly release.

The core approach involves taking a pre-trained latent video diffusion model, freezing its encoder/decoder components, and fine-tuning only the latent diffusion model backbone. The main methodological contribution lies in the temporal alignment of the input video frames, which they achieve using RoPE embedding interpolation.

The experiments presented in the paper include an ablation study to verify the effectiveness of their proposed temporal alignment approach. They also provide both qualitative and quantitative results (including a small user study) on their proposed benchmark. Finally, the authors compare their model against different styles of conditioning on the given frames.

**Strengths:**

- The problem of spatio-temporal video completion addresses a practically important and challenging task.
- A notable strength is the model's demonstrated capability for generating long-duration videos (up to approximately 70 seconds), as evidenced by the examples provided on the project webpage.
- The paper is well-written, clearly organized, and easy to follow.

**Weaknesses:**

The paper needs a more comprehensive experimental evaluation. Besides the ablation study, VideoCanvas is only compared to two other types of conditioning, and the evaluation is conducted solely on the authors' proposed benchmark. The paper would be significantly stronger with experiments that include comparisons against other video completion models and evaluation on existing video-completion benchmarks.

While the authors claim there are no other arbitrary spatio-temporal video completion benchmarks, the "Image Level" and "Video Level" tasks (in Table 2) only require full frames and align closely with prior video-completion and video interpolation tasks. In order to truly show the utility of the model, a comparison with more baselines on these existing task types is necessary.

Along the same lines, the paper misses some important, closely related, and well-established works such as [1, 2, 3, 4], to name a few. In particular, [2] was the first paper performing video completion from arbitrary frames and [3] performs video interpolation in latent diffusion. Not exactly what this paper does, but pretty closely related works. A comparison to these methods, either in the related work section or the experimental evaluation (or both), is necessary.

[1] Voleti, V., Jolicoeur-Martineau, A. and Pal, C. Mcvd: masked conditional video diffusion for prediction, generation, and interpolation. NeurIPS 2022.
[2] Harvey, W., Naderiparizi, S., Masrani, V., Weilbach, C. and Wood, F. Flexible diffusion modeling of long videos. NeurIPS 2022.
[3] Danier, D., Zhang, F. and Bull, D. Ldmvfi: Video frame interpolation with latent diffusion models. AAAI 2024.
[4] Jain, S., Watson, D., Tabellion, E., Poole, B. and Kontkanen, J. Video interpolation with diffusion models. CVPR 2024.

**Questions:**

- Is the proposed interpolation method specific to RoPE embedding? Is there anything stopping one from applying the same interpolation in sinusoidal embeddings?
- On line 396 and 397, the authors argue the reason that the reason Pixel-space Padding does not work well is the VAE was never trained on zero-filled inputs. However, VideoCanvas still uses the VAE for zero-filled inputs, when the inputs are patches of a frame. Why is this not a problem in VideoCanvas?
- In the experimetns (5.3 or 5.4), do the authors also fine-tune the diffusion model seperately for their baselines?
- In section 5.3, I expected "Latent-space Conditioning" to be an upper bound on the performance of the other models, since it is in some sense "cheating" by encoding the entire groundtruth video and providing the encoded latent, which encodes not only the given frame, but also the other N-1 frames in the same encoding window. Nonethelees, Figure 4 shows the PSNR for Latent-space Conditioning does not peak at the correct frame. Why is that?
- Do any of the reported metrics capture diversity of generated samples?

---

### Official Review · Reviewer_Xk9b · 2025-10-31

**Soundness:** 2
**Presentation:** 2
**Contribution:** 2
**Rating:** 2
**Confidence:** 3

**Summary:**

This paper proposes a framework VideoCanvas for addressing arbitrary spatio-temporal video completion task, and a hybrid conditioning strategies to decouple space and time: spatial control through zero-padding and temporal alignment by Temporal RoPE interpolation. Based on In-Context Conditioning Paradigm, VideoCanvas enables pixel-frame-aware control without retraining VAE and addresses the temporal ambiguity of causal VAEs. This paper also proposes VideoCanvasBench, which is the first benchmark that evaluates the spatio-temporal video completion.

**Strengths:**

1. It proposes the first benchmark, VideoCanvasBench to measure the spatio-temporal completion.
2. It might be easy to implement as all the techniques within the proposed method are from existing works.

**Weaknesses:**

1. I found the contribution of this paper is incremental: the training objective is diffusion/flow matching loss, the backbone is DiT, the time embedding is RoPE, and all the mechanisms the proposed method used are from existing works. The other contribution the paper mentions is VideoCanvasBench, which is the first benchmark to measure spatio-temporal completion but it is not a technical modelling facilitation.
2. Even though the temporal RoPE interpolation is from an existing work, but it would be better to give some math expressions.
3. The concern that the frozen VAE possibly processes an out-of-distribution inputs because of zero padding is mentioned, and the paper has a section as an ablation study about it, the proposed temporal RoPE interpolation still has zero padding in it and no further discussion about that. I am not quite convinced if the input is extremely sparse and what it would look like.
4. I am not quite familiar with this type of task, and all of the baselines used to compare to the proposed method is designed by the authors in the experiments. This weakens the evidence to show ICC could really work.

**Questions:**

1. Can you explain why the Pixelspace-Padding only changes the color of the image but it stays the same otherwise, i.e. the image is not distorted at all?
2. In Figure 6(a), it is commented ''Too Static' on Replace. and ''Unexpected Artifacts'' on Channel., but I do not really see the different on Frame 37 and Frame 40 across all the methods. In Figure 6(b), It seems that the generated sample by the proposed method shows two different creatures in Frame 37 and Frame 58 but it does not mean the transition is smooth and coherent. In addition, I wonder why those frames are selected for visualization? What is the rationale behind this for frame selection?

---

### Note · Authors · 2025-11-14

**Comment:**

We thank all reviewers for their valuable time, constructive feedback, and helpful suggestions.

**Withdrawal Confirmation:**

I have read and agree with the venue's withdrawal policy on behalf of myself and my co-authors.